# Sparse DNNs with Improved Adversarial Robustness

**Yiwen Guo[1,2]*    Chao Zhang[3]*    Changshui Zhang[2]    Yurong Chen[1]**
[1] Intel Labs China
[2] Institute for Artificial Intelligence, Tsinghua University (THUAI),
State Key Lab of Intelligent Technologies and Systems,
Beijing National Research Center for Information Science and Technology (BNRis),
Department of Automation, Tsinghua University
[3] Academy for Advanced Interdisciplinary Studies, Center for Data Science, Peking University
{yiwen.guo, yurong.chen}@intel.com    pkuzc@pku.edu.cn    zcs@mail.tsinghua.edu.cn

## Abstract

Deep neural networks (DNNs) are computationally/memory-intensive and vulnerable to adversarial attacks, making them prohibitive in some real-world applications. By converting dense models into sparse ones, pruning appears to be a promising solution to reducing the computation/memory cost. This paper studies classification models, especially DNN-based ones, to demonstrate that there exists intrinsic relationships between their sparsity and adversarial robustness. Our analyses reveal, both theoretically and empirically, that nonlinear DNN-based classifiers behave differently under $l_2$ attacks from some linear ones. We further demonstrate that an appropriately higher model sparsity implies better robustness of nonlinear DNNs, whereas over-sparsified models can be more difficult to resist adversarial examples.

## 1 Introduction

Although deep neural networks (DNNs) have advanced the state-of-the-art of many artificial intelligence techniques, some undesired properties may hinder them from being deployed in real-world applications. With continued proliferation of deep learning powered applications, one major concern raised recently is the heavy computation and storage burden that DNN models shall lay upon mobile platforms. Such burden stems from substantially redundant feature representations and parameterizations [6]. To address this issue and make DNNs less resource-intensive, a variety of solutions have been proposed. In particular, it has been reported that more than 90% of connections in a well-trained DNN can be removed using pruning strategies [14, 13, 28, 21, 23], while no accuracy loss is observed. Such a remarkable network sparsity leads to considerable compressions and speedups on both GPUs and CPUs [25]. Aside from being efficient, sparse representations are theoretically attractive [2, 8] and have made their way into tremendous applications over the past decade.

Orthogonal to the inefficiency issue, it has also been discovered that DNN models are vulnerable to adversarial examples—maliciously generated images which are perceptually similar to benign ones but can fool classifiers to make arbitrary predictions [26, 3]. Furthermore, generic regularizations (e.g., dropout and weight decay) do not really help on resisting adversarial attacks [11]. Such undesirable property may prohibit DNNs from being applied to security-sensitive applications. The cause of this phenomenon seems mysterious and remains to be an open question. One reasonable explanation is the local linearity of modern DNNs [11]. Quite a lot of attempts, including adversarial training [11, 27, 19], knowledge distillation [24], detecting and rejecting [18], and some gradient masking techniques like randomization [31], have been made to ameliorate this issue and defend adversarial attacks.

It is crucial to study potential relationships between the inefficiency (i.e., redundancy) and adversarial robustness of classifiers, in consideration of the inclination to avoid "robbing Peter to pay Paul", if possible. Towards shedding light on such relationships, especially for DNNs, we provide comprehensive analyses in this paper from both the theoretical and empirical perspectives. By introducing reasonable metrics, we reveal, somewhat surprising, that there is a discrepancy between the robustness of sparse linear classifiers and nonlinear DNNs, under $l_2$ attacks. Our results also demonstrate that an appropriately higher sparsity implies better robustness of nonlinear DNNs, whereas over-sparsified models can be more difficult to resist adversarial examples, under both the $l_\infty$ and $l_2$ circumstances.

## 2    Related Works

In light of the "Occam's razor" principle, we presume there exists intrinsic relationships between the sparsity and robustness of classifiers, and thus perform a comprehensive study in this paper. Our theoretical and empirical analyses shall cover both linear classifiers and nonlinear DNNs, in which the middle-layer activations and connection weights can all become sparse.

The (in)efficiency and robustness of DNNs have seldom been discussed together, especially from a theoretical point of view. Very recently, Gopalakrishnan et al. [12, 20] propose to sparsify the input representations as a defense and provide provable evidences on resisting $l_\infty$ attacks. Though intriguing, their theoretical analyses are limited to only linear and binary classification cases. Contemporaneous with our work, Wang et al. [29] and Ye et al. [32] experimentally discuss how pruning shall affect the robustness of some DNNs but surprisingly draw opposite conclusions. Galloway et al. [9] focus on binary DNNs instead of the sparse ones and show that the difficulty of performing adversarial attacks on binary networks DNNs remains as that of training.

To some extent, several very recent defense methods also utilize the sparsity of DNNs. For improved model robustness, Gao et al. [10] attempt to detect the feature activations exclusive to the adversarial examples and prune them away. Dhillon et al. [7] choose an alternative way that prunes activations stochastically to mask gradients. These methods focus only on the sparsity of middle-layer activations and pay little attention to the sparsity of connections.

## 3    Sparsity and Robustness of Classifiers

This paper aims at analyzing and exploring potential relationships between the sparsity and robustness of classifiers to untargeted white-box adversarial attacks, from both theoretical and practical perspectives. To be more specific, we consider models which learn parameterized mappings $\mathbf{x}_i \mapsto y_i$, when given a set of labelled training samples $\{(\mathbf{x}_i, y_i)\}$ for supervision. Similar to a bunch of other theoretical efforts, our analyses start from linear classifiers and will be generalized to nonlinear DNNs later in Section 3.2.

Generally, the sparsity of a DNN model can be considered in two aspects: the sparsity of connections among neurons and the sparsity of neuron activations. In particular, the sparsity of activations also include that of middle-layer activations and inputs, which can be treated as a special case. Knowing that the input sparsity has been previously discussed [12], we shall focus primarily on the weight and activation sparsity for nonlinear DNNs and just study the weight sparsity for linear models.

### 3.1    Linear Models

For simplicity of notation, we first give theoretical results for binary classifiers with $\hat{y}_i = \mathrm{sgn}(\mathbf{w}^T \mathbf{x}_i)$, in which $\mathbf{w}, \mathbf{x}_i \in \mathbb{R}^n$. We also ignore the bias term $b$ for clarity. Notice that $\mathbf{w}^T \mathbf{x} + b$ can be simply rewritten as $\hat{\mathbf{w}}^T [\mathbf{x}; 1]$ in which $\hat{\mathbf{w}} = [\mathbf{w}; b]$, so all our theoretical results in the sequel apply directly to linear cases with bias. Given ground-truth labels $y_i \in \{+1, -1\}$, a classifier can be effectively trained by minimizing some empirical loss $\sum_i \tau(-y_i \cdot \mathbf{w}^T \mathbf{x}_i)$ using a logistic sigmoid function like softplus: $\tau(\cdot) = \log(1 + \exp(\cdot))$ [11].

Adversarial attacks typically minimize an $l_p$ norm (e.g., $l_2$, $l_\infty$, $l_1$ and $l_0$) of the required perturbation under certain (box) constraints. Though not completely equivalent with the distinctions in our visual domain, such norms play a crucial role in evaluating adversarial robustness. We study both the $l_\infty$ and $l_2$ attacks in this paper. With an ambition to totalize them, we propose to evaluate the robustness

of linear models using the following metrics that describe the ability of resisting them respectively:

$$\textbf{Binary}: \quad r_\infty := \mathrm{E}_{\mathbf{x},y}\left(1_{y=\mathrm{sgn}(\mathbf{w}^T \check{\mathbf{x}})}\right),$$
$$r_2 := \mathrm{E}_{\mathbf{x},y}\left(1_{y=\hat{y}} \cdot d(\mathbf{x}, \tilde{\mathbf{x}})\right). \tag{1}$$

Here we let $\check{\mathbf{x}} = \mathbf{x} - \epsilon y \cdot \mathrm{sgn}(\mathbf{w})$ and $\tilde{\mathbf{x}} = \mathbf{x} - \mathbf{w}(\mathbf{w}^T\mathbf{x})/\|\mathbf{w}\|_2^2$ be the adversarial examples generated by applying the fast gradient sign (FGS) [11] and DeepFool [22] methods as representatives. Without box constraints on the image domain, they can be regarded as the optimal $l_\infty$ and $l_2$ attacks targeting on the linear classifiers [20, 22]. Function $d$ calculates the Euclidean distance between two $n$-dimensional vectors and we know that $d(\mathbf{x}, \tilde{\mathbf{x}}) = |\mathbf{w}^T\mathbf{x}|/\|\mathbf{w}\|_2$.

The introduced two metrics evaluate robustness of classifiers from two different perspectives: $r_\infty$ calculates the expected accuracy on (FGS) adversarial examples and $r_2$ measures a decision margin between benign examples from the two classes. For both of them, higher value indicates stronger adversarial robustness. Note that unlike some metrics calculating (maybe normalized) Euclidean distances between all pairs of benign and adversarial examples, our $r_2$ omits the originally misclassfied examples, which makes more sense if the classifiers are imperfect in the sense of prediction accuracy. We will refer to $\boldsymbol{\mu}_k := \mathrm{E}(\mathbf{x}|y = k, \hat{y} = k)$, which is the conditional expectation for class $k$.

Be aware that although there exists attack-agnostic guarantees on the model robustness [16, 30], they are all instance-specific. Instead of generalizing them to the entire input space for analysis, we focus on the proposed statistical metrics and present their connections to the guarantees later in Section 3.2. Some other experimentally feasible metrics shall be involved in Section 4. The following theorem sheds light on intrinsic relationships between the described robustness metrics and $l_p$ norms of $\mathbf{w}$.

**Theorem 3.1.** *(The sparsity and robustness of binary linear classifiers). Suppose that $\mathrm{P}_y(k) = 1/2$ for $k = \pm 1$, and an obtained linear classifier achieves the same expected accuracy $t$ on different classes, then we have*

$$r_2 = \frac{t}{2} \cdot \frac{\mathbf{w}^T(\boldsymbol{\mu}_{+1} - \boldsymbol{\mu}_{-1})}{\|\mathbf{w}\|_2} \quad and \quad r_\infty \leq \frac{t}{2} \cdot \frac{\mathbf{w}^T(\boldsymbol{\mu}_{+1} - \boldsymbol{\mu}_{-1})}{\epsilon\|\mathbf{w}\|_1}. \tag{2}$$

*Proof.* For $r_\infty$, we first rewrite it in the form of $\mathrm{Pr}(y \cdot \mathbf{w}^T\check{\mathbf{x}} > 0)$. We know from assumptions that $\mathrm{Pr}(\hat{y} = k|y = k) = t$ and $\mathrm{Pr}(y = k) = 1/2$, so we further get

$$r_\infty = \sum_{k=\pm 1} \frac{t}{2} \mathrm{Pr}\left(k \cdot \mathbf{w}^T\mathbf{x} > \epsilon\|\mathbf{w}\|_1 \,|\, y = k, \hat{y} = k\right), \tag{3}$$

by using the law of total probability and substituting $\check{\mathbf{x}}$ with $\mathbf{x} - \epsilon y \cdot \mathrm{sgn}(\mathbf{w})$. Lastly the result follows after using the Markov's inequality.

As for $r_2$, the proof is straightforward by similarly casting its definition into the sum of conditional expectations. That is,

$$r_2 = \sum_{k=\pm 1} \frac{t}{2} \mathrm{E}_{\mathbf{x}|y,\hat{y}}\left(\frac{|\mathbf{w}^T\mathbf{x}|}{\|\mathbf{w}\|_2} \,\Big|\, y = k, \hat{y} = k\right). \tag{4}$$

$\square$

Theorem 3.1 indicates clear relationships between the sparsity and robustness of linear models. In terms of $r_\infty$, optimizing the problem gives rise to a sparse solution of $\mathbf{w}$. By duality, maximizing the squared upper bound of $r_\infty$ also resembles solving a sparse PCA problem [5]. Reciprocally, we might also concur that a highly sparse $\mathbf{w}$ implies relatively robust classification results. Nevertheless, it seems that the defined $r_2$ has nothing to do with the sparsity of $\mathbf{w}$. It gets maximized iff $\mathbf{w}$ approaches $\boldsymbol{\mu}_{+1} - \boldsymbol{\mu}_{-1}$ or $\boldsymbol{\mu}_{-1} - \boldsymbol{\mu}_{+1}$, however, sparsifying $\mathbf{w}$ probably does not help on reaching this goal. In fact, under some assumptions about data distributions, the dense reference model can be nearly optimal in the sense of $r_2$. We will see this phenomenon remains in multi-class linear classifications in Theorem 3.2 but does not remain in nonlinear DNNs in Section 3.2. One can check Section 4.1 and 4.2 for some experimental discussions in more details.

Having realized that the $l_\infty$ robustness of binary linear classifiers is closely related to $\|\mathbf{w}\|_1$, we now turn to multi-class cases with the ground truth $y_i \in \{1, \ldots, c\}$ and prediction $\hat{y}_i = \arg\max_k(\mathbf{w}_k^T\mathbf{x}_i)$, in which $\mathbf{w}_k = W[:, k]$ indicates the $k$-th column of a matrix $W \in \mathbb{R}^{n \times c}$. Here the training objective

$f$ calculates the cross-entropy loss between ground truth labels and outputs of a softmax function. The introduced two metrics shall be slightly modified to:

$$\textbf{Multi-class}: \quad r_\infty := \mathrm{E}_{\mathbf{x},y}\left(1_{y=\arg\max_k(\mathbf{w}_k^T\tilde{\mathbf{x}})}\right),$$
$$r_2 := \mathrm{E}_{\mathbf{x},y}\left(1_{y=\hat{y}}\cdot d(\mathbf{x},\tilde{\mathbf{x}})\right). \tag{5}$$

Likewise, $\check{\mathbf{x}} = \mathbf{x} + \epsilon\cdot\mathrm{sgn}(\nabla f(\mathbf{x}))$ and $\tilde{\mathbf{x}} = \mathbf{x} - \mathbf{w}_\delta(\mathbf{w}_\delta^T\mathbf{x})/\|\mathbf{w}_\delta\|_2^2$ are the FGS and DeepFool adversarial examples under multi-class circumstances, in which $\mathbf{w}_\delta = \mathbf{w}_{\hat{y}} - \mathbf{w}_e$ and $e \in \{1,\ldots,c\} - \{\hat{y}\}$ is carefully chosen such that $|(\mathbf{w}_{\hat{y}} - \mathbf{w}_e)^T\mathbf{x}|/\|\mathbf{w}_{\hat{y}} - \mathbf{w}_e\|_2$ is minimized. Denote an averaged classifier by $\bar{\mathbf{w}} := \sum_k \mathbf{w}_k/c$, we provide upper bounds for both $r_\infty$ and $r_2$ in the following theorem.

**Theorem 3.2.** *(The sparsity and robustness of multi-class linear classifiers). Suppose that $\mathrm{P}_y(k) = 1/c$ for $k \in \{1,...,c\}$, and an obtained linear classifier achieves the same expected accuracy $t$ on different classes, then we have*

$$r_2 \le \frac{t}{c}\sum_{k=1}^c \frac{(\mathbf{w}_k - \bar{\mathbf{w}})^T\boldsymbol{\mu}_k}{\|\mathbf{w}_k - \bar{\mathbf{w}}\|_2} \quad and \quad r_\infty \le \frac{t}{c}\sum_{k=1}^c \frac{(\mathbf{w}_k - \bar{\mathbf{w}})^T\boldsymbol{\mu}_k}{\epsilon\|\mathbf{w}_k - \bar{\mathbf{w}}\|_1} \tag{6}$$

*under two additional assumptions: (I) FGS achieves higher per-class success rates than a weaker perturbation like $-\epsilon\cdot\mathrm{sgn}(\mathbf{w}_y - \bar{\mathbf{w}})$, (II) the FGS perturbation does not correct misclassifications.*

We present in Theorem 3.2 similar bounds for multi-class classifiers to that provided in Theorem 3.1, under some mild assumptions. Our proof is deferred to the supplementary material. We emphasize that the two additional assumptions are intuitively acceptable. First, increasing the classification loss in a more principled way, say using FGS, ought to diminish the expected accuracy more effectively. Second, with high probability, an original misclassification cannot be fixed using the FGS method, as one intends to do precisely the opposite.

Similarly, the presented bound for $r_\infty$ also implies sparsity, though it is the sparsity of $\mathbf{w}_k - \bar{\mathbf{w}}$. In fact, this is directly related with the sparsity of $\mathbf{w}_k$, considering that the classifiers can be post-processed to subtract their average simultaneously whilst the classification decision won't change for any possible input. Particularly, Theorem 3.2 also partially suits linear DNN-based classifications. Let the classifier $g_k$ be factorized in a form of $\mathbf{w}_k^T = (\mathbf{w}_k')^T W_{d-1}^T \ldots W_1^T$, it is evident to see that higher sparsity of the multipliers encourages higher probability of a sparse $\mathbf{w}_k$.

## 3.2 Deep Neural Networks

A nonlinear feedforward DNN is usually specified by a directed acyclic graph $G = (\mathcal{V}, \mathcal{E})$ [4] with a single root node for final outputs. According to the forward propagation rule, the activation value of each internal (and also output) node is calculated based on its incoming nodes and learnable weights corresponding to the edges. Nonlinear activation functions are incorporated to ensure the capacity. With biases, some nodes output a special value of one. We omit them for simplicity reasons as before.

Classifications are performed by comparing the prediction scores corresponding to different classes, which means $\hat{y} = \arg\max_{k\in\{1,...,c\}} g_k(\mathbf{x})$. Benefit from some very recent theoretic efforts [16, 30], we can directly utilize well-established robustness guarantees for nonlinear DNNs. Let us first denote by $B_p(\mathbf{x}, R)$ a close ball centred at $\mathbf{x}$ with radius $R$ and then denote by $L_{q,\mathbf{x}}^k$ the (best) local Lipschitz constant of function $g_{\hat{y}}(\mathbf{x}) - g_k(\mathbf{x})$ over a fixed $B_p(\mathbf{x}, R)$, if there exists one. It has been proven that the following lemma offers a reasonable lower bound for the required $l_p$ norm of instance-specific perturbations when all classifiers are Lipschitz continuous [30].

**Proposition 3.1.** [30] *Let $\hat{y} = \arg\max_{k\in\{1,...,c\}} g_k(\mathbf{x})$ and $\frac{1}{p} + \frac{1}{q} = 1$, then for any $\Delta_{\mathbf{x}} \in B_p(\mathbf{0}, R)$, $p \in \mathbb{R}^+$ and a set of Lipschitz continuous functions $\{g_k : \mathbb{R}^n \mapsto \mathbb{R}\}$, with*

$$\|\Delta_{\mathbf{x}}\|_p \le \min\left\{\min_{k\neq\hat{y}} \frac{g_{\hat{y}}(\mathbf{x}) - g_k(\mathbf{x})}{L_{q,\mathbf{x}}^k}, R\right\} := \gamma, \tag{7}$$

*it holds that $\hat{y} = \arg\max_{k\in\{1,...,c\}} g_k(\mathbf{x} + \Delta_{\mathbf{x}})$, which means the classification decision does not change on $B_p(\mathbf{x}, \gamma)$.*

Here the introduced $\gamma$ is basically an instance-specific lower bound that guarantees the robustness of multi-class classifiers. We shall later discuss its connections with our $r_p$s, for $p \in \{\infty, 2\}$, and now

we try providing a local Lipschitz constant (which may not be the smallest) of function $g_{\hat{y}}(\mathbf{x}) - g_k(\mathbf{x})$, to help us delve deeper into the robustness of nonlinear DNNs. Without loss of generality, we will let the following discussion be made under a fixed radius $R > 0$ and a given instance $\mathbf{x} \in \mathbb{R}^n$.

Some modern DNNs can be structurally very complex. Let us simply consider a multi-layer perceptron (MLP) parameterized by a series of weight matrices $W_1 \in \mathbb{R}^{n_0 \times n_1}, \ldots, W_d \in \mathbb{R}^{n_{d-1} \times n_d}$, in which $n_0 = n$ and $n_d = c$. Discussions about networks with more advanced architectures like convolutions, pooling and skip connections can be directly generalized [1]. Specifically, we have

$$g_k(\mathbf{x}_i) = \mathbf{w}_k^T \sigma(W_{d-1}^T \sigma(\ldots \sigma(W_1^T \mathbf{x}_i))), \tag{8}$$

in which $\mathbf{w}_k = W_d[:, k]$ and $\sigma$ is the nonlinear activation function. Here we mostly focus on "ReLU networks" with rectified-linear-flavoured nonlinearity, so the neuron activations in middle-layers are naturally sparse. Due to clarity reasons, we discuss the weight and activation sparsities separately. Mathematically, we let $\mathbf{a}_0 = \mathbf{x}$ and $\mathbf{a}_j = \sigma(W_j^T \mathbf{a}_{j-1})$ for $0 < j < d$ be the layer-wise activations. We will refer to

$$D_j(\mathbf{x}) := \mathrm{diag}\left(1_{W_j[:,1]^T \mathbf{a}_{j-1} > 0}, \ldots, 1_{W_j[:,n_j]^T \mathbf{a}_{j-1} > 0}\right), \tag{9}$$

which is a diagonal matrix whose entries taking value one correspond to nonzero activations within the $j$-th layer, and $M_j \in \{0,1\}^{n_{j-1} \times n_j}$, which is a binary mask corresponding to each (possibly sparse) $W_j$. Along with some analyses, the following lemma and theorem present intrinsic relationships between the adversarial robustness and (both weight and activation) sparsity of nonlinear DNNs.

**Lemma 3.1.** *(A local Lipschitz constant for ReLU networks). Let $\frac{1}{p} + \frac{1}{q} = 1$, then for any $\mathbf{x} \in \mathbb{R}^n$, $k \in \{1, \ldots, c\}$ and $q \in \{1, 2\}$, the local Lipschitz constant of function $g_{\hat{y}}(\mathbf{x}) - g_k(\mathbf{x})$ satisfies*

$$L_{q,\mathbf{x}}^k \leq \|\mathbf{w}_{\hat{y}} - \mathbf{w}_k\|_q \sup_{\mathbf{x}' \in B_p(\mathbf{x},R)} \prod_{j=1}^{d-1} \left(\|D_j(\mathbf{x}')\|_p \|W_j\|_p\right). \tag{10}$$

**Theorem 3.3.** *(The sparsity and robustness of nonlinear DNNs). Let the weight matrix be represented as $W_j = W_j' \circ M_j$, in which $\{M_j[u,v]\}$ are independent Bernoulli $B(1, 1 - \alpha_j)$ random variables and $0 \notin \{W_j'[u,v]\}$, for $j \in \{1, \ldots, d-1\}$. Then for any $\mathbf{x} \in \mathbb{R}^n$ and $k \in \{1, \ldots, c\}$, it holds that*

$$\mathrm{E}_{M_1,\ldots,M_{d-1}}(L_{2,\mathbf{x}}^k) \leq c_2 \cdot (1 - \eta(\alpha_1, \ldots, \alpha_{d-1}; \mathbf{x})) \tag{11}$$

*and*

$$\mathrm{E}_{M_1,\ldots,M_{d-1}}(L_{1,\mathbf{x}}^k) \leq c_1 \cdot (1 - \eta(\alpha_1, \ldots, \alpha_{d-1}; \mathbf{x})), \tag{12}$$

*in which function $\eta$ is monotonically increasing w.r.t. each $\alpha_j$, $c_2 = \|\mathbf{w}_{\hat{y}} - \mathbf{w}_k\|_2 \prod_j \|W_j'\|_F$ and $c_1 = \|\mathbf{w}_{\hat{y}} - \mathbf{w}_k\|_1 \prod_j \|W_j'\|_{1,\infty}$ are two constants.*

*Proof Sketch.* Function $\prod \|D_j(\cdot)\|_p \|W_j\|_p$ defined on $\mathbb{R}^n$ is bounded from above and below, thus we know there exists an $\hat{\mathbf{x}} \in B_p(\mathbf{x}, R)$ satisfying

$$L_{q,\mathbf{x}}^k \leq \|\mathbf{w}_{\hat{y}} - \mathbf{w}_k\|_q \prod_j \|D_j(\hat{\mathbf{x}})\|_p \|W_j\|_p. \tag{13}$$

Particularly, $\prod \|D_j(\hat{\mathbf{x}})\|_p \neq 0$ is fulfilled iff $\|D_{d-1}(\hat{\mathbf{x}})\|_p \neq 0$ (i.e., it equals 1 for $q \in \{1, 2\}$). Under the assumptions on $M_j$, we know that the entries of $W_j$ are independent of each other, thus

$$\begin{aligned}
\mathrm{Pr}_{M_1,\ldots,M_{d-1}}(D_{d-1}(\hat{\mathbf{x}})[u,u] = 0) &= \mathrm{Pr}_{M_1,\ldots,M_{d-1}}(W_{d-1}[:,u]^T \mathbf{a}_{d-2} \leq 0) \\
&\geq \prod_{u'} (\alpha_{d-1} + \xi_{d-2,u'} - \alpha_{d-1}\xi_{d-2,u'}),
\end{aligned} \tag{14}$$

in which $\xi_{d-2,u'}$ is a newly introduced scalar that equals or less equals to the probability of the $u'$-th neuron being deactivated. In this manner, we can recursively define the function $\eta$ and it is easy to validate its monotonicity. Additionally, we prove that $c_q \geq \|\mathbf{w}_{\hat{y}} - \mathbf{w}_k\|_q \mathrm{E}\left(\prod \|W_j\|_p \|D_{d-1}(\hat{\mathbf{x}})\|_p = 1\right)$ holds for $q \in \{1, 2\}$ and the result follows. See the supplementary material for a detailed proof. $\square$

In Lemma 3.1 we introduce probably smaller local Lipschitz constants than the commonly known ones (i.e., $c_2$ and $c_1$), and subsequently in Theorem 3.3 we build theoretical relationships between $L_{q,\mathbf{x}}^k$ and the network sparsity, for $q \in \{1, 2\}$ (i.e., $p \in \{\infty, 2\}$). Apparently, $L_{q,\mathbf{x}}^k$ is prone to get

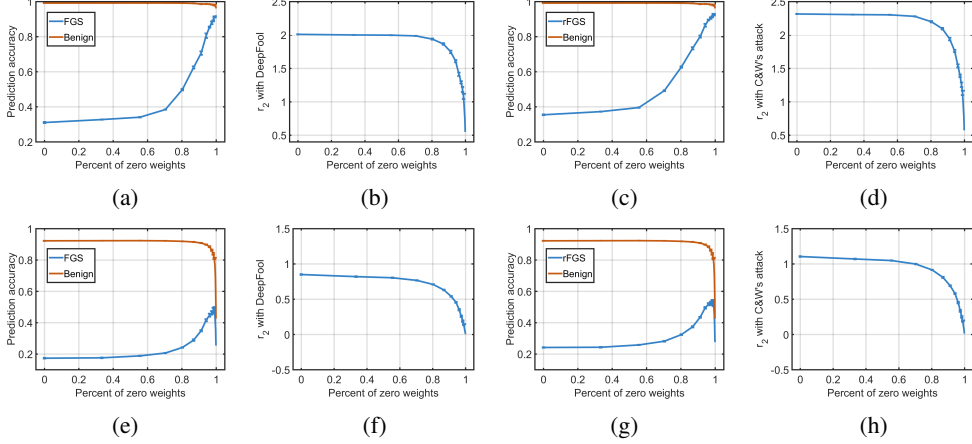

Figure 1: The robustness of linear classifiers with varying weight sparsity. Upper: binary classification between "1"s and "7"s, lower: multi-class classification on the whole MNIST test set.

smaller if any weight matrix gets more sparse. It is worthy noting that the local Lipschitz constant is of great importance in evaluating the robustness of DNNs, and it is effective to regularize DNNs by just minimizing $L_{q,\mathbf{x}}^k$, or equivalently $\|\nabla g_{\hat{y}}(\mathbf{x}) - \nabla g_k(\mathbf{x})\|_q$ for differentiable continuous functions [16]. Thus we reckon, when the network is over-parameterized, an appropriately higher weight sparsity implies a larger $\gamma$ and stronger robustness. There are similar conclusions if $\mathbf{a}_j$ gets more sparse.

Recall that in the linear binary case, we apply the DeepFool adversarial example $\tilde{\mathbf{x}}$ when evaluating the robustness using $r_2$. It is not difficult to validate that the equality $d(\mathbf{x}, \tilde{\mathbf{x}}) = |(\mathbf{w}_{\hat{y}} - \mathbf{w}_{k \neq \hat{y}})^T \mathbf{x}| / L_{2,\mathbf{x}}^k$ holds for such $\tilde{\mathbf{x}}$ and $\mathbf{w}_{\pm 1} := \pm \mathbf{w}$, which means the DeepFool perturbation ideally minimizes the Euclidean norm and helps us measure a lower bound in this regard. This can be directly generalized to multi-class classifiers. Unlike $r_2$ which represents a margin, our $r_\infty$ is basically an expected accuracy. Nevertheless, we also know that a perturbation of $-\epsilon y \cdot \text{sgn}(\mathbf{w})$ shall successfully fool the classifiers if $\epsilon \geq |(\mathbf{w}_{\hat{y}} - \mathbf{w}_{k \neq \hat{y}})^T \mathbf{x}| / L_{1,\mathbf{x}}^k$.

# 4 Experimental Results

In this section, we conduct experiments to testify our theoretical results. To be consistent, we still start from linear models and turn to nonlinear DNNs afterwards. As previously discussed, we perform both $l_\infty$ and $l_2$ attacks on the classifiers to evaluate their adversarial robustness. In addition to the FGS [11] and DeepFool [22] attacks which have been thoroughly discussed in Section 3, we introduce two more attacks in this section for extensive comparisons of the model robustness.

**Adversarial attacks.** We use the FGS and randomized FGS (rFGS) [27] methods to perform $l_\infty$ attacks. As a famous $l_\infty$ attack, FGS has been widely exploited in the literature. In order to generate adversarial examples, it calculates the gradient of training loss w.r.t. benign inputs and uses its sign as perturbations, in an element-wise manner. The rFGS attack is a computationally efficient alternative to multi-step $l_\infty$ attacks with an ability of breaking adversarial training-based defences. We keep its hyper-parameters fixed for all experiments in this paper. For $l_2$ attacks, we choose DeepFool and the C&W's attack [3]. DeepFool linearises nonlinear classifiers locally and approximates the optimal perturbations iteratively. C&W's method casts the problem of constructing adversarial examples as optimizing an objective function without constraints, such that some recent gradient-descent-based solvers can be adopted. On the base of different attacks, four $r_2$ and $r_\infty$ values can be calculated for each classification model.

## 4.1 The Sparse Linear Classifier Behaves Differently under $l_\infty$ and $l_2$ Attacks

In our experiments on linear classifiers, both the binary and multi-class scenarios shall be evaluated. We choose the well-established MNIST dataset as a benchmark, which consists of 70,000 $28 \times 28$ images of handwritten digits. According to the official test protocol, 10,000 of them should be used for performance evaluation and the remaining 60,000 for training. For experiments on the binary

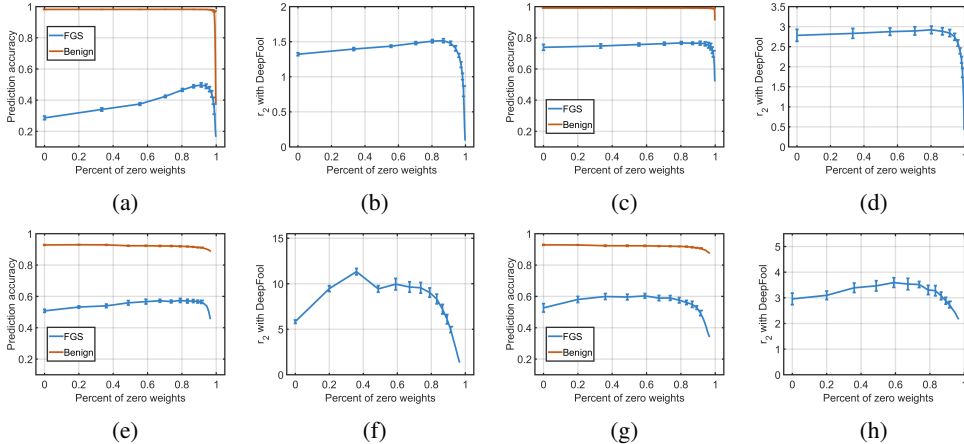

Figure 2: The robustness of nonlinear DNNs with varying weight sparsity. (a)-(b): LeNet-300-100, (c)-(d): LeNet-5, (e)-(f): the VGG-like network, (g)-(h): ResNet-32.

cases, we randomly choose a pair of digits (e.g., "0" and "8" or "1" and "7") as positive and negative classes. Linear classifiers are trained following our previous discussions and utilizing the softplus function: $\min_{\mathbf{w},b} \sum_i \log(1 + \exp(-y_i(\mathbf{w}^T\mathbf{x}_i + b)))$. Parameters $\mathbf{w}$ and $b$ are randomly initialized and learnt by means of stochastic gradient descent with momentum. For the "1" and "7" classification case, we train 10 reference models from different initializations and achieve a prediction accuracy of $99.17 \pm 0.00\%$ on the benign test set. For the classification of all 10 classes, we train 10 references similarly and achieve a test-set accuracy of $92.26 \pm 0.08\%$.

To produce models with different weight sparsities, we use a progressive pruning strategy [14]. That being said, we follow a pipeline of iteratively pruning and re-training. Within each iteration, a portion ($\rho$) of nonzero entries of $\mathbf{w}$, whose magnitudes are relatively small in comparison with the others, will be directly set to zero and shall never be activated again. After $m$ times of such "pruning", we shall collect $10(m + 1)$ models from all 10 dense references. Here we set $m = 16, \rho = 1/3$ so the achieved final percentage of zero weights should be $99.74\% \approx 1 - (1 - \rho)^m$. We calculate the prediction accuracies on adversarial examples (i.e., $r_\infty$) under different $l_\infty$ attacks and the average Euclidean norm of required perturbations (i.e., $r_2$) under different $l_2$ attacks to evaluate the adversarial robustness of different models in practice. For $l_\infty$ attacks, we set $\epsilon = 0.1$.

Figure 1 illustrates how our metrics of robustness vary with the weight sparsity. We only demonstrate the variability of the first 12 points (from left to right) on each curve, to make the bars more resolvable. The upper and lower subfigures correspond to binary and multi-class cases, respectively. Obviously, the experimental results are consistent with our previous theoretical ones. While sparse linear models are prone to be more robust in the sense of $r_\infty$, their $r_2$ robustness maintains similar or becomes even slightly weaker than the dense references, until there emerges inevitable accuracy degradations on benign examples (i.e., when $r_\infty$ may drop as well). We also observe from Figure 1 that, in both the binary and multi-class cases, $r_2$ starts decreasing much earlier than the benign-set accuracy. Though very slight in the binary case, the degradation of $r_2$ actually occurs after the first round of pruning (from $2.0103 \pm 0.0022$ to $2.0009 \pm 0.0016$ with DeepFool incorporated, and from $2.3151 \pm 0.0023$ to $2.3061 \pm 0.0023$ with the C&W's attack).

## 4.2 Sparse Nonlinear DNNs Can be Consistently More Robust

Regarding nonlinear DNNs, we follow the same experimental pipeline as described in Section 4.1. We train MLPs with 2 hidden fully-connected layers and convolutional networks with 2 convolutional layers, 2 pooling layers and 2 fully-connected layers as references on MNIST, following the "LeNet-300-100" and "LeNet-5" architectures in network compression papers [14, 13, 28, 21]. We also follow the training policy suggested by Caffe [17] and train network models for 50,000 iterations with a batch size of 64 such that the training cross-entropy loss does not decrease any longer. The well-trained reference models achieve much higher prediction accuracies (LeNet-300-100: $98.20 \pm 0.07\%$ and LeNet-5: $99.11 \pm 0.04\%$) than previous tested linear ones on the benign test set.

**Weight sparsity.** Then we prune the dense references and illustrate some major results regarding the robustness and weight sparsity in Figure 2 (a)-(d). (See Figure 3 in our supplementary material for results under rFGS and the C&W's attack.) Weight matrices/tensors within each layer is uniformly pruned so the network sparsity should be approximately equal to the layer-wise sparsity. As expected, we observe similar results to previous linear cases in the context of our $r_\infty$ but significantly different results in $r_2$. Unlike previous linear models which behave differently under $l_\infty$ and $l_2$ attacks, nonlinear DNN models show a consistent trend of adversarial robustness with respect to the sparsity. In particular, we observe increased $r_\infty$ and $r_2$ values under different attacks when continually pruning the models, until the sparsity reaches some thresholds and leads to inevitable capacity degradations. For additional verifications, we calculate the CLEVER [30] scores that approximate attack-agnostic lower bounds of the $l_p$ norms of required perturbations (in Table 3 in the supplementary material).

Experiments are also conducted on CIFAR-10, in which deeper nonlinear networks can be involved. We train 10 VGG-like network models [23] (each incorporates 12 convolutional layers and 2 fully-connected layers) and 10 ResNet models [15] (each incorporates 31 convolutional layers and a single fully-connected layers) from scratch. Such deep architectures lead to average prediction accuracies of 93.01% and 92.89%. Still, we prune dense network models in the progressive manner and illustrate quantitative relationships between the robustness and weight sparsity in Figure 2 (e)-(h). The first and last layers in each network are kept dense to avoid early accuracy degradation on the benign set. The same observations can be made. Note that the ResNets are capable of resisting some DeepFool examples, for which the second and subsequent iterations make little sense and can be disregarded.

**Activation sparsity.** Having testified relationship between the robustness and weight sparsity of nonlinear DNNs, we now examine the activation sparsity. As previously mentioned, the middle-layer activations of ReLU incorporated DNNs are naturally sparse. We simply add a $l_1$ norm regularization of weight matrices/tensors to the learning objective to encourage higher sparsities and calculate $r_\infty$ and $r_2$ accordingly. Experiments are conducted on MNIST. Table 1 summarizes the results, in which "Sparsity (**a**)" indicates the percentage of deactivated (i.e., zero) neurons feeding to the last fully-connected layer. Here the $r_\infty$ and $r_2$ values are calculated using the FGS and DeepFool attacks, respectively. Apparently, we still observe positive correlations between the robustness and (activation) sparsity in a certain range.

Table 1: The robustness of DNNs regularized using the $l_1$ norm of weight matrices/tensors.

| Network | $r_\infty$ | $r_2$ | Accuracy | Sparsity (**a**) |
|---|---|---|---|---|
| | 0.2862±0.0113 | 1.3213±0.0207 | 98.20±0.07% | 45.25±1.14% |
| LeNet-300-100 | **0.3993±0.0079** | **1.5903±0.0240** | **98.27±0.04%** | 75.92±0.54% |
| | 0.2098±0.0133 | 1.1440±0.0402 | 97.96±0.07% | 95.22±0.18% |
| | 0.7388±0.0188 | 2.7831±0.1490 | 99.11±0.04% | 51.26±1.88% |
| LeNet-5 | **0.7729±0.0081** | **3.1688±0.1203** | **99.19±0.05%** | 97.54±0.10% |
| | 0.6741±0.0162 | 2.0799±0.0522 | 99.10±0.06% | 99.64±0.02% |

### 4.3 Avoid "Over-pruning"

We discover from Figure 2 that the sharp decrease of the adversarial robustness, especially in the sense of $r_2$, may occur in advance of the benign-set accuracy degradation. Hence, it can be necessary to evaluate the adversarial robustness of DNNs during an aggressive surgery, even though the prediction accuracy of compressed models may remain competitive with their references on benign test-sets. To further explore this, we collect some off-the-shelf sparse models (including a $56\times$ compressed LeNet-300-100 and a $108\times$ compressed LeNet-5) [13] and their corresponding dense references from the Internet and hereby evaluate their $r_\infty$ and $r_2$ robustness. Table 2 compares the robustness of different models. Obviously, these extremely sparse models are more vulnerable to the DeepFool attack, and what's worse, the over $100\times$ pruned LeNet-5 seems also more vulnerable to FGS, which suggests researchers to take care and avoid "over-pruning" if possible. One might also discover the fact with other pruning methods.

Table 2: The robustness of pre-compressed nonlinear DNNs and their provided dense references.

| Model | $r_\infty$ | $r_2$ | Sparsity ($W$) |
|---|---|---|---|
| LeNet-300-100 dense | 0.2663 | **1.3899** | 0.00% |
| LeNet-300-100 sparse | **0.3823** | 1.1058 | 98.21% |
| LeNet-5 dense | **0.7887** | **2.7226** | 0.00% |
| LeNet-5 sparse | 0.6791 | 1.7383 | 99.07% |

## 5  Conclusions

In this paper, we study some intrinsic relationships between the adversarial robustness and the sparsity of classifiers, both theoretically and empirically. By introducing plausible metrics, we demonstrate that unlike some linear models which behave differently under $l_\infty$ and $l_2$ attacks, sparse nonlinear DNNs can be consistently more robust to both of them than their corresponding dense references, until their sparsity reaches certain thresholds and inevitably causes harm to the network capacity. Our results also demonstrate that such sparsity, including sparse connections and middle-layer neuron activations, can be effectively imposed using network pruning and $l_1$ regularization of weight tensors.

## Acknowledgement

We would like to thank anonymous reviewers for their constructive suggestions. Changshui Zhang is supported by NSFC (Grant No. 61876095, No. 61751308 and No. 61473167) and Beijing Natural Science Foundation (Grant No. L172037).

## Footnotes

*The first two authors contributed equally to this work.

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
