[Supplementary Material · supp.pdf]

# Supplementary Material

**Yiwen Guo**[1,2]*  **Chao Zhang**[3]*  **Changshui Zhang**[2]  **Yurong Chen**[1]
[1] Intel Labs China  [2] Tsinghua University  [3] Peking University
{yiwen.guo, yurong.chen}@intel.com  pkuzc@pku.edu.cn  zcs@mail.tsinghua.edu.cn

## A  Proof of Theorem 3.2

*Proof.* Recall that we directly have $\Pr(\hat{y} = k | y = k) = t$ and $\Pr(y = k) = 1/c$, then according to the additional assumption (II) and the law of total probability, it holds that

$$r_\infty = \frac{t}{c} \sum_{k=1}^{c} \Pr\left(\mathbf{w}_k^T \check{\mathbf{x}} > \max_{l \neq k} \mathbf{w}_l^T \check{\mathbf{x}} \,|\, y = k, \hat{y} = k\right). \tag{1}$$

Let us denote $\acute{\mathbf{x}} := \mathbf{x} - \epsilon \cdot \mathrm{sgn}(\mathbf{w}_y - \bar{\mathbf{w}})$ as the degraded adversarial example, then we further have

$$
\begin{aligned}
r_\infty &\leq \frac{t}{c} \sum_{k=1}^{c} \Pr\left(\mathbf{w}_k^T \acute{\mathbf{x}} > \max_{l \neq k} \mathbf{w}_l^T \acute{\mathbf{x}} \,|\, y = k, \hat{y} = k\right). \\
&\leq \frac{t}{c} \sum_{k=1}^{c} \Pr\left(\mathbf{w}_k^T \acute{\mathbf{x}} > (c\bar{\mathbf{w}} - \mathbf{w}_k)^T \acute{\mathbf{x}}/(c-1) \,|\, y = k, \hat{y} = k\right) \\
&= \frac{t}{c} \sum_{k=1}^{c} \Pr\left((\mathbf{w}_k - \bar{\mathbf{w}})^T \mathbf{x} > \epsilon \|\mathbf{w}_k - \bar{\mathbf{w}}\|_1 \,|\, y = k, \hat{y} = k\right),
\end{aligned}
\tag{2}
$$

by taking advantage of the assumption (I) and replacing the max operation with an average. Finally our result follows after using the Markov's inequality.

The method for deriving the upper bound of $r_2$ is analogous to our proof of Theorem 3.1. We consider the conditional expectations to get:

$$
\begin{aligned}
r_2 &= \sum_{k=1}^{c} \frac{t}{c} \mathrm{E}_{\mathbf{x}|y,\hat{y}} \left( \min_{l \neq k} \frac{|(\mathbf{w}_k - \mathbf{w}_l)^T \mathbf{x}|}{\|\mathbf{w}_k - \mathbf{w}_l\|_2} \,\Big|\, y = k, \hat{y} = k \right) \\
&\leq \sum_{k=1}^{c} \frac{t}{c} \mathrm{E}_{\mathbf{x}|y,\hat{y}} \left( \frac{|(\mathbf{w}_k - \bar{\mathbf{w}})^T \mathbf{x}|}{\|\mathbf{w}_k - \bar{\mathbf{w}}\|_2} \,\Big|\, y = k, \hat{y} = k \right) \\
&= \frac{t}{c} \sum_{k=1}^{c} \frac{(\mathbf{w}_k - \bar{\mathbf{w}})^T \boldsymbol{\mu}_k}{\|\mathbf{w}_k - \bar{\mathbf{w}}\|_2},
\end{aligned}
\tag{3}
$$

in which the following inequality is used:

$$
\begin{aligned}
\min_{l \neq k} \frac{|(\mathbf{w}_k - \mathbf{w}_l)^T \mathbf{x}|}{\|\mathbf{w}_k - \mathbf{w}_l\|_2} &\leq 1 \Bigg/ \left( \sum_{l \neq k} \frac{\|\mathbf{w}_k - \mathbf{w}_l\|_2}{(\mathbf{w}_k - \mathbf{w}_l)^T \mathbf{x}} \cdot \frac{(\mathbf{w}_k - \mathbf{w}_l)^T \mathbf{x}}{|(\mathbf{w}_k - \bar{\mathbf{w}})^T \mathbf{x}|c} \right) \\
&= 1 \Bigg/ \left( \sum_{l \neq k} \frac{\|\mathbf{w}_k - \mathbf{w}_l\|_2}{|(\mathbf{w}_k - \bar{\mathbf{w}})^T \mathbf{x}|c} \right) \\
&\leq \frac{|(\mathbf{w}_k - \bar{\mathbf{w}})^T \mathbf{x}|}{\|\mathbf{w}_k - \bar{\mathbf{w}}\|_2}.
\end{aligned}
\tag{4}
$$

$\square$

## B    Proof of Lemma 3.1

**Lemma B.1.** *Let $\sigma(\cdot) = \max\{\cdot, 0\}$ be a rectifier function, then for any $x_a, x_b \in \mathbb{R}$, it holds that*

$$|\sigma(x_a) - \sigma(x_b)| \leq \max\{1_{x_a>0}, 1_{x_b>0}\}|x_a - x_b|. \tag{5}$$

The proof is self-evident. We generalize Lemma B.1 to $n$-dimensional Euclidean space as follows.

**Lemma B.2.** *Let $\sigma(\cdot)$ be a function employing element-wise rectifier, then for any $\mathbf{x}_a, \mathbf{x}_b \in \mathbb{R}^n$, it holds that*

$$\|\sigma(\mathbf{x}_a) - \sigma(\mathbf{x}_b)\|_p \leq \|H(\mathbf{x}_a, \mathbf{x}_b)\|_p \|\mathbf{x}_a - \mathbf{x}_b\|_p, \tag{6}$$

*in which $H(\mathbf{x}_a, \mathbf{x}_b) := \mathrm{diag}(\max\{1_{\mathbf{x}_a[1]>0}, 1_{\mathbf{x}_b[1]>0}\}, \ldots, \max\{1_{\mathbf{x}_a[n]>0}, 1_{\mathbf{x}_b[n]>0}\}).$*

*Proof.* From definition of the vector $l_p$ norms, we get

$$
\begin{aligned}
\|\sigma(\mathbf{x}_a) - \sigma(\mathbf{x}_b)\|_p &= (\textstyle\sum_u |\sigma(\mathbf{x}_a[u]) - \sigma(\mathbf{x}_b[u])|^p)^{\frac{1}{p}} \\
&\leq (\textstyle\sum_u \max\{1_{\mathbf{x}_a[u]>0}, 1_{\mathbf{x}_b[u]>0}\}|\mathbf{x}_a[u] - \mathbf{x}_b[u]|^p)^{\frac{1}{p}} \\
&= \|H(\mathbf{x}_a, \mathbf{x}_b)(\mathbf{x}_a - \mathbf{x}_b)\|_p \\
&\leq \|H(\mathbf{x}_a, \mathbf{x}_b)\|_p \|(\mathbf{x}_a - \mathbf{x}_b)\|_p,
\end{aligned} \tag{7}
$$

in which the last inequality is obtained using the definition of the induced matrix norms. $\square$

**Lemma B.3.** *Given $\mathbf{x}_a \in \mathbb{R}^n$, there exists $\epsilon \in \mathbb{R}^+$, for all $\mathbf{x}_b$ satisfying $\|\mathbf{x}_a - \mathbf{x}_b\|_\infty < \epsilon$, the diagonal matrix $H(\mathbf{x}_a, \mathbf{x}_b)$ defined in Lemma B.2 can be simplified as $H(\mathbf{x}_b) := \mathrm{diag}(1_{\mathbf{x}_b[1]>0,\ldots,\mathbf{x}_b[n]>0}).$*

*Proof.* Let us first prove a one-dimensional form. Given $x_a \neq 0$, we can let $\epsilon = \frac{|x_a|}{2}$, then for all $x_b$ satisfying $|x_b - x_a| < \epsilon$, it holds that $\max\{1_{x_a>0}, 1_{x_b>0}\} = 1_{x_a>0} = 1_{x_b>0}$. If given $x_a = 0$, then $1_{x_a>0} = 0$ and $1_{x_a>0} \leq 1_{x_b>0}$ for all $x_b \in \mathbb{R}$. Hence, we have $\max\{1_{x_a>0}, 1_{x_b>0}\} = 1_{x_b>0}$ and $\max\{1_{x_a>0}, 1_{x_b>0}\}|x_a - x_b| = 1_{x_b>0}|x_a - x_b|$. Above derivations can be directly generalized to higher dimensions, following (7). $\square$

Finally it comes to our formal proof of Lemma 3.1. We provide two ways of proving it as below.

*Proof.* Let us first denote $h_k(\cdot) := g_{\hat{y}}(\cdot) - g_k(\cdot)$ and

$$
\begin{aligned}
D_j(\mathbf{x}', \mathbf{x}) := \mathrm{diag}\Big( &\max\{1_{W_j[:,1]^T \mathbf{a}'_{j-1}>0}, 1_{W_j[:,1]^T \mathbf{a}_{j-1}>0}\}, \ldots, \\
&\max\{1_{W_j[:,n_j]^T \mathbf{a}'_{j-1}>0}, 1_{W_j[:,n_j]^T \mathbf{a}_{j-1}>0}\}\Big).
\end{aligned} \tag{8}
$$

According to our Lemma B.2, it follows that,

$$
\begin{aligned}
&|h_k(\mathbf{x}') - h_k(\mathbf{x})| \\
\leq &\|\mathbf{w}_{\hat{y}} - \mathbf{w}_k\|_q \|\sigma(W_{d-1}^T \sigma(\ldots \sigma(W_1^T \mathbf{x}'))) - \sigma(W_{d-1}^T \sigma(\ldots \sigma(W_1^T \mathbf{x})))\|_p \\
\leq &\|\mathbf{w}_{\hat{y}} - \mathbf{w}_k\|_q \|D_{d-1}(\mathbf{x}', \mathbf{x})\|_p \|W_{d-1}\|_p \|\sigma(\ldots \sigma(W_1^T \mathbf{x}'))) - \sigma(\ldots \sigma(W_1^T \mathbf{x})))\|_p \\
\leq &\|\mathbf{w}_{\hat{y}} - \mathbf{w}_k\|_q \max\{\|D_{d-1}(\mathbf{x}')\|_p, \|D_{d-1}(\mathbf{x})\|_p\}\|W_{d-1}\|_p \|\sigma(\ldots \sigma(W_1^T \mathbf{x}'))) - \sigma(\ldots \sigma(W_1^T \mathbf{x})))\|_p \\
&\cdots \\
\leq &\left( \|\mathbf{w}_{\hat{y}} - \mathbf{w}_k\|_q \prod_{j=1}^{d-1} \max\{\|D_j(\mathbf{x}')\|_p, \|D_j(\mathbf{x})\|_p\}\|W_j\|_p \right) \|\mathbf{x}' - \mathbf{x}\|_p,
\end{aligned}
$$

for any $\mathbf{x}' \in B_p(\mathbf{x}, R)$, in which the Hölder's inequality is used and all matrix norms are the induced norms. In fact, since $D_j(\mathbf{x}')$ is diagonal and its entries belong to $\{0, 1\}$, function $\prod \|D_j(\cdot)\|_p$, for $q \in \{1, 2\}$, has at most 2 possible values (i.e., 0 and 1). Hence, we further have

$$
\begin{aligned}
|h_k(\mathbf{x}') - h_k(\mathbf{x})| &\leq \|\mathbf{x}' - \mathbf{x}\|_p \|\mathbf{w}_{\hat{y}} - \mathbf{w}_k\|_q \sup_{\mathbf{x}'' \in B(\mathbf{x}, R)} \prod_{j=1}^{d-1} (\max\{\|D_j(\mathbf{x}'')\|_p, \|D_j(\mathbf{x})\|_p\}\|W_j\|_p) \\
&= \|\mathbf{x}' - \mathbf{x}\|_p \|\mathbf{w}_{\hat{y}} - \mathbf{w}_k\|_q \sup_{\mathbf{x}'' \in B(\mathbf{x}, R)} \prod_{j=1}^{d-1} (\|D_j(\mathbf{x}'')\|_p \|W_j\|_p),
\end{aligned}
$$

by exploiting the fact that $\mathbf{x} \in B_p(\mathbf{x}, R)$. [2] The Lemma is now evident from what we have proved.

We also provide an alternative proof, in which the Lemma 3.3 in Weng et al.'s paper [3] is utilized. Denote $D^+ h_k(\mathbf{x}; \mathbf{d}) := \lim_{t \to 0^+} \frac{h_k(\mathbf{x}+t\mathbf{d}) - h_k(\mathbf{x})}{t}$, it follows that

$$L^k_{q,\mathbf{x}} \leq \sup_{\mathbf{x}' \in S} \{ | \sup_{\|\mathbf{d}\|_p = 1} D^+ h_k(\mathbf{x}'; \mathbf{d}) | \}, \tag{9}$$

in which $S$ is a convex bounded closed set and $\mathbf{x} \in S$. According to Lemma B.3, it follows that

$$|h_k(\mathbf{x}' + t\mathbf{d}) - h_k(\mathbf{x}')|/t$$
$$\leq \|\mathbf{w}_{\hat{y}} - \mathbf{w}_k\|_q \|\sigma(W_{d-1}^T \sigma(\ldots \sigma(W_1^T(\mathbf{x}' + t\mathbf{d})))) - \sigma(W_{d-1}^T \sigma(\ldots \sigma(W_1^T \mathbf{x}')))\|_p / t$$
$$\leq \|\mathbf{w}_{\hat{y}} - \mathbf{w}_k\|_q \|D_{d-1}(\mathbf{x}' + t\mathbf{d})\|_p \|W_{d-1}\|_p \|\sigma(\ldots \sigma(W_1^T(\mathbf{x}' + t\mathbf{d}))) - \sigma(\ldots \sigma(W_1^T \mathbf{x}'))\|_p / t$$
$$\cdots$$
$$\leq \left( \|\mathbf{w}_{\hat{y}} - \mathbf{w}_k\|_q \prod_{j=1}^{d-1} \|D_j(\mathbf{x}' + t\mathbf{d})\|_p \|W_j\|_p \right) \|\mathbf{d}\|_p,$$

when $t \to 0^+$, in which the Hölder's inequality is used as well. Therefore, if we let $S = B_p(\mathbf{x}, R)$, it holds that $\mathbf{x}' + t\mathbf{d} \in B_p(\mathbf{x}, R)$ (when $t \to 0^+$) and

$$L^k_{q,\mathbf{x}} \leq \|\mathbf{w}_{\hat{y}} - \mathbf{w}_k\|_q \sup_{\mathbf{x}' \in B_p(\mathbf{x}, R)} \prod_{j=1}^{d-1} (\|D_j(\mathbf{x}')\|_p \|W_j\|_p). \tag{10}$$

$\square$

## C  Proof and Discussions of Theorem 3.3

*Proof.* We know from our Lemma 3.1 that there must exist at least an $\hat{\mathbf{x}} \in B_p(\mathbf{x}, R)$ such that,

$$0 \leq L^k_{q,\mathbf{x}} \leq \|\mathbf{w}_{\hat{y}} - \mathbf{w}_k\|_q \prod_{j=1}^{d-1} (\|D_j(\hat{\mathbf{x}})\|_p \|W_j\|_p), \tag{11}$$

as the function $\prod (\|D_j(\cdot)\|_p \|W_j\|_p)$ is bounded from above and below for any $\mathbf{x} \in \mathbb{R}^n$, $q \in \{1, 2\}$ and $k \in \{1, \ldots, c\}$. Let us first consider the $q = 2$ case. For simplicity of notation, we will ignore the subscript $M_1, \cdots, M_i$ within probability and expectation terms in the sequel. It holds that

$$E(L^k_{2,\mathbf{x}}) \leq E \left( \|\mathbf{w}_{\hat{y}} - \mathbf{w}_k\|_2 \prod_{j=1}^{d-1} \|D_j(\hat{\mathbf{x}})\|_2 \|W_j\|_2 \right)$$
$$= \|\mathbf{w}_{\hat{y}} - \mathbf{w}_k\|_2 E \left( \prod_{j=1}^{d-1} \|D_j(\hat{\mathbf{x}})\|_2 \|W_j\|_2 \right) \tag{12}$$
$$= \|\mathbf{w}_{\hat{y}} - \mathbf{w}_k\|_2 \Pr(\|D_{d-1}(\hat{\mathbf{x}})\|_2 = 1) \prod_{j=1}^{d-1} E(\|W_j\|_2 | \|D_{d-1}(\hat{\mathbf{x}})\|_2 = 1).$$

Based on our assumptions on $M_j$, it is not difficult to show that $\{D_j(\hat{\mathbf{x}})[u, u]\}$ are independent Bernoulli distributed random variables. Let us denote $D_j(\hat{\mathbf{x}})[u, u] \sim B(1, 1 - \beta_{j,u})$, in which $\beta_{j,u}$ should rely only on $\mathbf{x}$ and maybe $\{W_j\}$, then it is easy to validate that

$$\Pr(\|D_{d-1}(\hat{\mathbf{x}})\|_2 = 1) = 1 - \prod_{u=1}^{n_{d-1}} \beta_{d-1,u}. \tag{13}$$

For the $j$-th layer, we also have

$$
\begin{aligned}
\mathrm{E}\left(\|W_j\|_2|\|D_{d-1}(\hat{\mathbf{x}})\|_2 = 1\right) \leq & \mathrm{E}\left(\|W_j\|_F|\|D_{d-1}(\hat{\mathbf{x}})\|_2 = 1\right) \\
\leq & \mathrm{E}\left(\|W_j'\|_F|\|D_{d-1}(\hat{\mathbf{x}})\|_2 = 1\right) \\
= & \|W_j'\|_F,
\end{aligned}
\tag{14}
$$

in which $\|\cdot\|_F$ indicates the Frobenius norm. Consequently, it holds that

$$
\begin{aligned}
\mathrm{E}(L_{2,\mathbf{x}}^k) \leq & c_2 \cdot \mathrm{Pr}\left(\|D_{d-1}(\hat{\mathbf{x}})\|_2 = 1\right) \\
\leq & c_2 \cdot (1 - \eta(\alpha_1, \ldots, \alpha_{d-1}; \mathbf{x})),
\end{aligned}
\tag{15}
$$

in which the function $\eta(\alpha_1, \ldots, \alpha_{d-1}; \mathbf{x})$ is recursively defined, with a range of $[0, 1]$. First, it is defined that:

$$
\eta(\alpha_1, \ldots, \alpha_{d-1}; \mathbf{x}) = \prod_u \xi_{d-1,u}.
\tag{16}
$$

Second, for $j \in \{2, \ldots, d-1\}$, it is further defined that

$$
\xi_{j,u} = \prod_{u'}(\alpha_j + \xi_{j-1,u'} - \alpha_j \xi_{j-1,u'}),
\tag{17}
$$

for $u \in \{1, \ldots, n_j\}$. Third, we define $\xi_{1,u} = (\alpha_1)^{\hat{n}_0}$ for $u \in \{1, \ldots, n_1\}$, in which $\hat{n}_0$ indicates the number of nonzero pixels in $\hat{\mathbf{x}}$. It's easy to prove that $\xi_{j,u} \leq \beta_{j,u}$ holds and thus the result follows. We can also validate that $\eta(\alpha_1, \cdots, \alpha_{d-1}; \mathbf{x})$ is monotonically increasing (i.e., $1 - \eta(\alpha_1, \cdots, \alpha_{d-1}; \mathbf{x})$ is monotonically decreasing) w.r.t. each $\alpha_j$, by using the chain rule.

The $q = 1$ case is proven similarly, except for

$$
\begin{aligned}
\mathrm{E}(\|W_j\|_\infty|\|D_j(\hat{\mathbf{x}})\|_\infty = 1) \leq & \mathrm{E}\left(\|W_j\|_\infty|\|D_j(\hat{\mathbf{x}})\|_\infty = 1\right) \\
\leq & \mathrm{E}\left(\|W_j'\|_\infty|\|D_j(\hat{\mathbf{x}})\|_\infty = 1\right) \\
= & \|W_j'\|_\infty.
\end{aligned}
\tag{18}
$$

The induced matrix norm $\|\cdot\|_\infty$ can be rewritten as the group norm $\|\cdot\|_{1,\infty}$. $\qquad\square$

On the base of our introduced (probably smaller) local Lipschitz constants (than the commonly known ones, i.e., $c_2$ and $c_1$), we build theoretical relationships between the robustness and network sparsity. It is worthy noting that such constants are of great importance in evaluating the robustness of DNNs, and it is effective to regularize DNNs by just minimizing $L_{q,\mathbf{x}}^k$, or equivalently $\|\nabla g_{\hat{y}}(\mathbf{x}) - \nabla g_k(\mathbf{x})\|_q$ for differentiable continuous functions [2]. In particular, if the network model is over-parameterized and redundant, pruning may impose little effect on the value of $g_{\hat{y}}(\mathbf{x}) - g_k(\mathbf{x})$. Thus we reckon, for such DNN models, an appropriately higher weight sparsity implies a larger value of $\gamma$ and thus further stronger robustness. Definitely, over-sparsifying shall lead to a significant decrease in $g_{\hat{y}}(\mathbf{x}) - g_k(\mathbf{x})$, and the robustness is by no means assured. We also note that if the network model is less redundant, a reasonably small sparsity may suffice to lead to a decrease in $g_{\hat{y}}(\mathbf{x}) - g_k(\mathbf{x})$ and a complex effect to the adversarial robustness, which may also be considered as "over-sparsifying". There are similar conclusions if $\mathbf{a}_j$ gets sparser instead (i.e., when $\beta_{j,u}$ gets larger).

## D  More Experimental Results

Due to the length limit of our paper, some experimental results are illustrated here. First we show in Figure 3, for nonlinear DNNs, how $r_\infty$ and $r_2$ calculated using rFGS and the C&W's attack vary with the weight sparsity. The results are basically consistent with those on the base of FGS and DeepFool. Note that the $r_2$ in Figure 3 (d) starts decreasing since the third round of pruning instead of the very beginning. It is attractive to test the C&W's attack also on the VGG-like network and ResNet models. However, it seems computationally more expensive than the other tested attacks, and unfortunately we were not able to get its results for all $10(m+1)$ models with limited computational resources.

One might also be curious about the robustness and sparsity of linear DNNs. Following the architecture of "LeNet-300-100", we construct a linear MLP without ReLU activations and test the impact of the model sparsity on the adversarial robustness on MNIST. We set $m = 16$ and illustrate the results in Figure 4. The linear MLP models are more vulnerable to the tested $l_\infty$ and $l_2$ attacks than

their "one layer counterparts" as tested in Section 4.1, possibly on account of the training strategy. As expected, we observe a considerable increase in $r_\infty$ when the weight sparsity is high ($\geq 80\%$). Apart from this, it is a little bit surprising that there exists a very shallow "valley" on each blue curve, between the points corresponding to the dense references and some extremely sparse models, which is worth exploring in future works.

(a)  (b)  (c)  (d)

(e)  (f)

Figure 3: The robustness of nonlinear DNNs with varying weight sparsity. Specifically, (a)-(b): LeNet-300-100, (c)-(d): LeNet-5, (e): VGG-like network, (f): ResNet-32.

(a)  (b)  (c)  (d)

Figure 4: The robustness of linear MLPs with varying weight sparsity.

We also report experimental results in regard of the CLEVER scores in Table 3. The four LeNet-5 models correspond to the dense reference and three sparse models after 3, 5 and 15 times of network pruning, respectively. We observe similar trend to that illustrated in Figure 2, in the sense of $l_\infty$ and $l_2$. Results in the $l_1$ norm are different from the other two, which is worth exploring in the future. Contemporaneous with our work, some other general metrics are also proposed (e.g., [1]). It can be interesting to further evaluate the adversarial robustness of sparse models with them.

Table 3: The CLEVER scores of dense and sparse nonlinear DNNs.

| Model | $l_\infty$ | $l_2$ | $l_1$ | Sparsity ($W$) |
|---|---|---|---|---|
| LeNet-300-100 | $0.0383\pm0.0006$ | $0.7595\pm0.0112$ | $\mathbf{3.8515\pm0.0668}$ | 0.00% |
| | $\mathbf{0.0489\pm0.0006}$ | $\mathbf{0.8075\pm0.0098}$ | $3.8055\pm0.0644$ | 83.19% |
| | $0.0356\pm0.0038$ | $0.3085\pm0.0255$ | $1.1611\pm0.1303$ | 99.53% |
| LeNet-5 | $0.0552\pm0.0019$ | $1.0281\pm0.0313$ | $\mathbf{4.3718\pm0.1383}$ | 0.00% |
| | $\mathbf{0.0577\pm0.0017}$ | $\mathbf{1.0360\pm0.0226}$ | $4.0236\pm0.1315$ | 65.70% |
| | $\mathbf{0.0577\pm0.0016}$ | $0.9994\pm0.0318$ | $3.5609\pm0.1507$ | 83.19% |
| | $0.0277\pm0.0020$ | $0.2974\pm0.0314$ | $0.8295\pm0.1189$ | 99.53% |

## E  Imbalance Classes

For simplicity of notations, we assume the samples are generated from unbiased distributions in Theorem 3.1 and 3.2. Here we note that the obtained theoretical results generalize to imbalance data

as well. Take the binary case (i.e., Theorem 3.1) as an example, suppose $P_y(k) = p_k$ (instead of 1/2) for $k = \pm 1$, then following the same line of derivation we will have:

$$r_2 = t \cdot \frac{\mathbf{w}^T(p_{+1}\boldsymbol{\mu}_{+1} - p_{-1}\boldsymbol{\mu}_{-1})}{\|\mathbf{w}\|_2} \quad \text{and} \quad r_\infty \leq t \cdot \frac{\mathbf{w}^T(p_{+1}\boldsymbol{\mu}_{+1} - p_{-1}\boldsymbol{\mu}_{-1})}{\epsilon\|\mathbf{w}\|_1}. \tag{19}$$

## Footnotes

*The first two authors contributed equally to this work.

[2] It can easily be seen that $\sup \prod \max\{\|D_j(\mathbf{x}'')\|_p, \|D_j(\mathbf{x})\|_p\} = \sup \prod \|D_j(\mathbf{x}'')\|_p$ holds for $q \in \{1, 2\}$. Definitely, $\sup \prod \max\{\|D_j(\mathbf{x}'')\|_p, \|D_j(\mathbf{x})\|_p\} \geq \sup \prod \|D_j(\mathbf{x}'')\|_p$. Thus if the equation does not hold, it must be $\sup \prod \max\{\|D_j(\mathbf{x}'')\|_p, \|D_j(\mathbf{x})\|_p\} = 1$ and $\sup \prod \|D_j(\mathbf{x}'')\|_p = 0$. Further, it follows with simple deductions that $\sup \max\{\|D_{d-1}(\mathbf{x}'')\|_p, \|D_{d-1}(\mathbf{x})\|_p\} = 1$ and $\sup \|D_{d-1}(\mathbf{x}'')\|_p = 0$, for $q \in \{1, 2\}$ (i.e., $p \in \{\infty, 2\}$), and this contradicts the fact that $\mathbf{x} \in B_p(\mathbf{x}, R)$.