[Reviews · NeurIPS 2018]

Reviewer 1



The paper studies the connection between sparsity of linear classifier as well as deep neural networks and their robustness against adversarial attacks. Based on the two new introduced metrics, the authors show that sparse deep neural network are more robust against l_2 and l_inf attacks compared to their dense counterparts. Interestingly, in the linear case the behavior is differently, where sparsity helps against l_inf attacks but it seems that dense models are more robust against l_2 attacks. I think the paper offers some interesting new insights both theoretically and empirically into adversarial robustness of deep neural networks and how it is connected to sparsity. Even though it is maybe not surprising that sparsity helps to make models more robust, given Occam's razor, the paper sheds a new light in the case of adversarial attacks. I do not have any major comments, however, to allow for a better understanding of the paper, the authors could clarify the following points: 1) The author only consider pruning to make model sparse. How would the empirical insights translate to other methods for sparsity, such as l2 or l1 regularization? 2) How are the results connected to the bias-variance-tradeoff? 3) Do the insights also translate to regressions?

Reviewer 2



This paper assumes an intrinsic relationship between the sparsity and the robustness of DNN models. The authors prove that this assumption holds for linear models under the $l_infinity$ attacker model, and it also holds for nonlinear models under both $l_2$ and $l_infinity$ attacker models. Then, the authors demonstrate some experimental results which are consistent with the theory. In general, this paper is well-written, and it is a plausible investigation on the issues of DNN robustness. The problem studied in this paper is very important and also challenging. The methods of the paper are inspiring. However, there are also critical issues in this paper, which jeopardize the importance of the achieved results. So I tend to give a weak reject to this paper. More details are shown below. Critical Issues: 1) The authors adopt two metrics to reflect the robustness of DNNs. However, they have not shown how general are such measurements. 2) When proving Theorem 3.1 and Theorem 3.2, the authors assume that the labels of the dataset follow uniform distributions. But they have not discussed whether the theorems hold for the scenarios of unbalanced datasets. In practice, it is very often that some labels occur more frequently than others. If the theorems cannot hold in such cases, the importance of the results would be affected. 3) The proof of nonlinear DNNs does not consider pooling functions, which are popular in present neural networks. Besides, the authors obtain two bounds in the proof, but these bounds are not reflected in the experimental results. Moreover, the robustness of Figure 2 drops dramatically after a threshold, which confuses the reviewer about the correctness of the proof. More Issues: 1) Page 2 line 51: the related papers published in the ICLR workshop this year draw conclusions opposite to this paper. However, the authors did not address the difference between the related papers and this paper. Since these results appear conflicting to each other, this reviewer is confused about the correctness of this paper if there is no appropriate explanation. 2) Page 3 line 93: the authors claim that the $r_2$ metric makes more sense if the classifiers are less accurate. However, the question is whether it is worth improving the robustness if the classifier is inaccurate itself? 3) Page 3 line 114: the authors conjecture that the robustness of linear models is unrelated to the sparsity with respect to $r_2$. This conclusion further confuses this reviewer about the generality of the two metrics. Minor Issues: Page 2 line 52: “shows” => “show” Page 6 line 245: The bar above 0.3 seems a typo. Page 7 line 263: The description of the DNNs is not clear. Page 7 line 264: “2 convolutional layer” => “2 convolutional layers”

Reviewer 3



This paper studied the relationship between sparsity and adversarial robustness for several classification models (linear and DNN based). Both theoretical and empirical results are provided to reveal the different behavior of linear model and DNN under l2 attacks and sparseness increase the robustness of DNN in general. All in all, the paper is well organized, and the empirical experiments are sufficient. I did not go through the detail proof of the theorem, but the sketch looks valid. Detail comments and question are as follows. - It seems the two proposed metrics and analysis are only for l2 and deepfool type of attacks. Can the result and conclusion have been extended to other adversarial attacks. Such as L-BFGS, BIM, Jacobian-based Saliency Map Attack. - It would be nice to discuss the connection between proposed evalution metrics and the recently proposed rebustness metric used to measure DNNs, such as Noise Sensitivity Score and skewness proposed in this paper https://arxiv.org/pdf/1806.01477.pdf - In the experiment part, it would be nice to show the result under different \epsilson for L_infinity attack to show empirically the conclusion still hold for a larger pertubation. Minor comments. 1. The notation of \hat{x} and \tilde{x} in define r_2 and r_infinity look much alike. It may be better to change to other notation to avoid confusion. 2. Captions in figure 1 and figure 2 to explain of FGS and Benign will help reader better understanding the results. **** I have read the author's rebuttal and it clarify some of the detail. And I will keep my score.